# HARDMATH: A Benchmark Dataset for Challenging Problems in Applied Mathematics

**Jingxuan Fan,**[*] **Sarah Martinson,**[*] **Erik Y. Wang,**[*] **Kaylie Hausknecht,**[*]
**Jonah Brenner,  Danxian Liu,  Nianli Peng,  Corey Wang,  Michael P. Brenner**
School of Engineering and Applied Sciences
Harvard University
Cambridge, MA 02138, USA

## Abstract

Advanced applied mathematics problems are not well-represented in existing benchmarking datasets used to evaluate Large Language Models (LLMs). To address this, we introduce **HARDMATH**, the Harvard Approximate Reasoning Dataset for Mathematics—a dataset of 1,466 difficult problems inspired by Harvard University's graduate course on asymptotic methods. The dataset contains a diverse set of challenging applied mathematics problems with worked solutions that employ various analytical approximation methods. Developing such solutions typically requires multiple modes of analysis—including mathematical reasoning, the use of computational tools, and subjective judgment—making this a challenging problem for LLMs. We establish a framework that auto-generates an arbitrarily large number of 'hard' applied mathematics problems with approximate analytical solutions that include validity checks against numerical ground-truths. We evaluate frontier LLMs on **HARDMATH-MINI**, a sub-sampled test set of 366 problems, as well as on 40 word problems formulated in applied science contexts. Even leading closed-source models like GPT-4 achieve only 43.8% overall accuracy with few-shot Chain-of-Thought prompting, and all models demonstrate significantly lower performance compared to results on existing mathematics benchmark datasets. We additionally conduct a detailed error analysis to gain insights into the failure cases of LLMs. These results demonstrate limitations of current LLM performance on advanced graduate-level asymptotic math problems and underscore the importance of datasets like **HARDMATH** to advance mathematical abilities of LLMs.

*Keywords*    approximation, asymptotic analysis, benchmark dataset, LLM evaluation,

## 1   Introduction

Many scientific and engineering problems involve mathematical equations, such as integrals, ordinary differential equations (ODEs), and partial differential equations (PDEs), that rarely have closed-form solutions. Traditional mathematics courses and most Large Language Model (LLM) benchmark datasets focus on problems with exact, analytical solutions. However, these benchmarks overlook a large class of math problems often arising in applied sciences that require *approximate* solutions, which are essential for gaining insights into complex systems. Numerical solutions to such problems can be useful, but they lack the explanatory power offered by approximate analytical methods, e.g. asymptotic and applied analysis.

---

[*]Equal contribution
All data and code used for this paper can be found at: https://github.com/sarahmart/HARDMath.

38th Conference on Neural Information Processing Systems (NeurIPS 2024).

To address this gap, we introduce **HARDMATH**, the Harvard Approximate Reasoning Dataset for Mathematics. This benchmark dataset is designed to evaluate LLMs on their ability to solve applied mathematics problems that require approximation techniques. **HARDMATH** contains 1,466 problems inspired by Harvard University's graduate course on asymptotic methods; it covers polynomials, ODEs, and integrals that often arise in real scientific and engineering contexts but that cannot be solved exactly. The dataset emphasizes problems that require advanced mathematical reasoning and approximations, offering a more challenging and diverse testbed for LLMs compared to existing datasets, which mostly focus on simpler, symbolically solvable calculations [1, 2, 3, 4].

Rather than sourcing problems from textbooks or standardized tests, we develop a codebase for automatically generating problems and step-by-step solutions. Our dataset includes a larger set for fine-tuning and two test sets for evaluating LLMs' mathematical reasoning on approximation methods. Here, we evaluate the accuracy of LLMs on our dataset and study their common error modes. We find that current LLMs perform poorly overall on these problems and demonstrate significant room for improvement.

## 2 Related work

### 2.1 Mathematics Datasets

Most mathematics datasets for evaluating or training LLMs focus on elementary arithmetic or word problems. Notable examples include **MATH** (12,500 high school competition-style problems) [3], **GSM8K** (8,500 multistep grade-school problems) [4], and **ODYSSEY-MATH** (387 hand-curated problems across various difficulty levels) [5]. While these datasets are valuable for assessing basic LLM math performance, most are limited in scope and complexity.

Recent efforts targeting more advanced problems are often manually sourced. Datasets like **JEEBENCH** [6] and a subset of **MATHBENCH** [1] include some college-level topics, such as ODEs and multivariable calculus. **GHOSTS** includes more advanced problems from graduate-level texts on functional analysis, topology, and probability theory [7], while **ARB** features formal math problems from qualifying exams at Harvard and Berkeley [8]. However, these datasets often (1) are limited in size and scalability, (2) focus on formal mathematics, or (3) cull problems from textbooks protected by copyrights. Notably, none of the existing datasets (Table 1) focus on advanced applied mathematics. **HARDMATH** fills this gap by presenting a large corpus of problems that require approximation techniques from asymptotics to be solved. **HARDMATH** is also highly scalable with a codebase for data generation. Since these problems cannot be formalized using tools like Lean or solved with symbolic computation software, they present the ideal domain for evaluating how LLMs integrate natural language reasoning and code-based tools to solve out-of-training sample math problems.

Table 1: Comparison of **HARDMATH** with related datasets. Note that for all datasets excluding **MATH**, we report the number of relevant problems at a comparable difficulty to our dataset (e.g., **THEORY-KNOWLEDGE-COLLEGE** in **MATHBENCH**, and **GRAD-TEXT** and **HOLES-IN-PROOFS** from **GHOSTS**.) **HARDMATH** is the largest graduate-level dataset.

| Dataset | Size | Data Generation | Difficulty |
|---|---|---|---|
| **MATH** [3] | 12.5K | Manual | High School |
| **MATHBENCH-T** [1] | 632 | Manual, Algorithmic | Undergraduate |
| **JEEBENCH** [6] | 236 | Manual | High School |
| **GHOSTS** [7] | 190 | Manual | Graduate |
| **ARB** [8] | 34 | Manual | Graduate |
| **HARDMATH** (Ours) | 1.4K | Algorithmic | Graduate |

### 2.2 Recent interest in advanced mathematics reasoning

As LLMs continue to improve, there has been growing interest in developing more challenging benchmarks. A notable example is the recent open challenge, *Humanity's Last Exam*, which aims to create the world's most difficult public AI benchmark, requesting questions that "only exceptional

individuals can answer correctly" and do not involve "straightforward calculation/computation" [9]. Similarly, frontier models have been advancing quickly, and many are explicitly focused on quantitative and scientific reasoning, such as OpenAI's recent o1 series. In line with our motivation for developing **HARDMATH** to better track the progress of LLMs, OpenAI argues that "recent frontier models do so well on **MATH** and **GSM8K** that these benchmarks are no longer effective at differentiating models" [10].

## 3   Datasets

**HARDMATH** contains four problem classes with seven distinct problem types covering nondimensionalization, polynomial root-finding, ODEs, and integrals, as well as 40 handwritten word problems designed to place the problems in applied scientific contexts (see Appendix A.1 for problem details). The main dataset (1,060 problems) is suitable for model development, while the **HARDMATH-mini** evaluation set (366 problems) is used for benchmarking LLM performance. Fig. 1 shows a breakdown of the datasets by problem type.

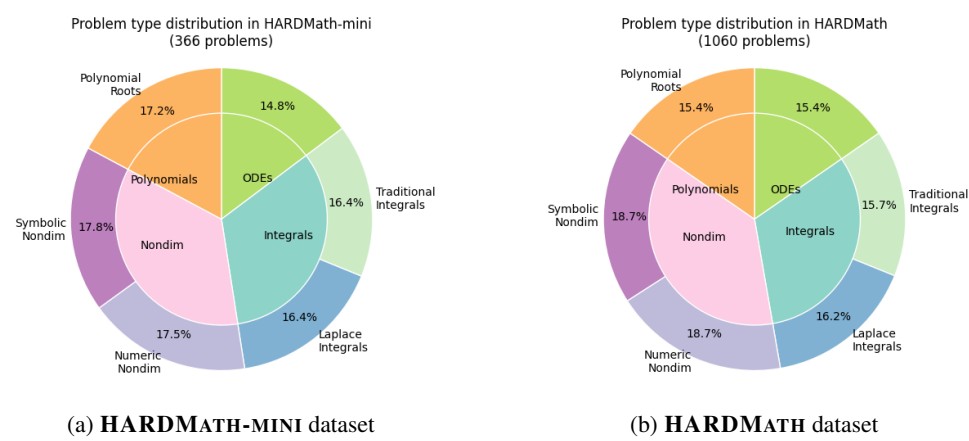

(a) **HARDMATH-MINI** dataset          (b) **HARDMATH** dataset

Figure 1: Breakdowns of the **HARDMATH-MINI** (left) and the **HARDMATH** (right) datasets.

Solutions to all **HARDMATH** problems share a common reasoning framework; the *Method of Dominant Balance* simplifies problems by focusing on terms that 'dominate' the solution's behavior and can significantly simplify the equation [11]. Solution methods also involve combining sophisticated computational and analytical techniques, such as self-consistency checks and the use of numerical methods. To solve these problems, subjective decisions about solution regimes to consider, terms to include, and approximation methods must be made with rigorous justification, which is challenging for current LLMs.

Implementation of this reasoning framework is realized through a robust data generation process. The data generation code uses `SymPy` [12] and `SciPy` [13] to implement mathematical procedures for generating approximate analytical solutions tailored to each problem class. Problems are generated randomly by combining sets of random coefficients, functional forms, and initial conditions. Solutions are generated algorithmically, with key steps described in explanatory texts. The main results are embedded in the LaTeX `\boxed{}` command, following conventions from other mathematics datasets (e.g. **MATH** [3]). Each problem type includes: 1) LaTeX-formatted problem statements, 2) LaTeX-formatted solution steps, 3) accuracy demonstrations comparing analytical and numerical solutions, and 4) metadata descriptors of the problem and solution types (Appendix A.1).

We evaluate solutions by calculating the relative error between analytical and numerical results at selected evaluation points. Problems are included in the dataset only if their solutions are within 10% of the numerical ground-truth, ensuring that all problems in **HARDMATH** maintain high accuracy. For polynomial root correction problems, we further check that the corrections improve on the original approximation.

# 4 Evaluation

## 4.1 Model choice and evaluation protocols

We evaluate several leading LLMs on **HARDMATH-MINI**, a subset of 366 problems representative of **HARDMATH** (Fig. 1). Closed-source LLMs evaluated include GPT-3.5 [14, 15, 16], GPT-4 [17] and o1-mini [18], open-source LLMs include Llama3 [19] and CodeLlama [20]. All models are tested in zero- and few-shot settings with Chain-of-Thought (CoT) prompting, which encourages complex reasoning capabilities by providing intermediate steps in sample answers [21]. Prompts and hyper-parameters are detailed in Appendix A.3.4.

We focus our evaluation on the four key problem types in **HARDMATH**: *Nondim* (symbolic and numerical nondimensionalization), *Roots* (polynomial root-finding), *ODEs* (nonlinear ODEs), and *Integrals* (traditional and Laplace integrals). Models are evaluated for accuracy and common error modes using zero- and few-shot CoT prompting. Prompts contain example question-solution pairs, problem setup and formatting hints (Appendix A.3.1). Following Hendrycks et al. [3], automatic assessment compares the final model-generated answer (A.3.1) to the true solution (both in LaTeX \boxed{} environments), using SymPy-based [12] equivalence checks and numerical evaluations. We also develop a procedural grading system using GPT-4o to (1) provide intermediate step grading for multi-step solutions, and (2) assess models' ability to make approximation judgments, which allow for a range of self-consistent solutions. Rubrics are adapted from grading guidelines of the course that inspired the **HARDMATH** problems (Appendix A.3.2 ). Human grading on a subset of LLM responses shows good alignment with GPT-based grading, with average score adjustments summarized in Appendix A.3.3.

## 4.2 Model performance and error mode analysis

Here, we report the accuracy of models across problem types and prompting settings (Table 2, Appendix A.4, Fig. 3). Few-shot CoT prompting enhances model performance across the board, particularly for o1-mini and GPT-4, which demonstrate the most substantial improvements, consistent with findings from [21] (Fig. 3a). The performance increase associated with prompting varies by problem-type; gains tend to saturate quickly on more challenging problems such as *ODEs*(Appendix A.4, Fig. 4). Notably, OpenAI's new o1-mini, though with much smaller parameter size, outperforms other models at all tested shot levels, confirming its optimization for STEM reasoning [18].

o1-mini with 5-shot CoT achieves the highest overall accuracy of 62.3%, while Llama3-8b achieves 20.2%, the highest among open-source models. In contrast, Llama3-8b performs significantly better on existing datasets, achieving 30.0% on **MATH** (4-shot CoT) and 79.6% on **GSM-8K** (8-shot CoT)[19]), compared to its 20.2% on **HARDMATH-MINI**. GPT-4 also shows strong performance on **MATH** (72.2%, 0-shot CoT), **GSM-8K** (92.0%, 5-shot CoT) [22, 17], and a recently released advanced mathematical dataset **MINIGHOSTS** (average score of 4.15 out of 5). Yet, GPT-4 achieves only 43.8% on **HARDMATH-MINI**. Similarly, o1-mini demonstrates 90.0% accuracy on **MATH-500** with 0-shot CoT [18], but achieves only 62.3% accuracy on **HARDMath-mini** with 5-shot CoT. This reveals a significant performance increase compared to other models on some (e.g. *Nondim*) but not all problem types. This suggests that the **HARDMATH** benchmark presents problems that remain challenging and unfamiliar to even the most performant LLMs developed for advanced STEM reasoning.

We also evaluate model responses across varying levels of correctness, allowing us to identify common error patterns. When breaking down performance by correct, partially correct, and incorrect responses, we observe that few-shot prompting improves performance to different degrees across problem types (Fig. 2). LLM solutions to harder problems, like *ODEs* and *Integrals*, are rarely fully correct, but receive more partial credit with increasing CoT shots. In contrast, for simpler problems like *Roots*, advanced models (o1-mini and GPT-4) get more fully correct responses with increasing CoT shots (Fig. 2, 5). Fig. 6 compares GPT-4's responses at 0 vs. 5 shot CoT on *Roots*, showing that the most common error mode—incorrectly setting up dominant balances—gets significantly reduced. Instead, errors shift to more nuanced issues, such as missing cases or failing to calculate complex roots (examples in Appendix A.4.2). This indicates that CoT improves the model's application of dominant balance techniques, enabling it to overcome simple mistakes.

Table 2: Evaluation Accuracy (percentage) on the **HARDMATH** evaluation set.

| Model | ALL | Nondim | Roots | ODEs | Integrals |
|---|---|---|---|---|---|
| **Closed-source models** | | | | | |
| GPT-3.5 (0 shot) | 6.04 | 5.05 | 17.2 | 1.39 | 3.33 |
| GPT-3.5 (1 shot CoT) | 14.2 | 6.11 | 29.3 | 6.94 | 18.2 |
| GPT-3.5 (5 shot CoT) | 24.6 | 24.3 | 35.0 | 16.2 | 23.1 |
| GPT-4 (0 shot) | 14.0 | 6.04 | 33.7 | 7.87 | 14.9 |
| GPT-4 (1 shot CoT) | 37.6 | 36.5 | 52.8 | 15.9 | 40.5 |
| GPT-4 (5 shot CoT) | 43.8 | 48.6 | 57.3 | 21.7 | 41.4 |
| o1-mini (0 shot CoT) | 29.8 | 38.1 | 24.3 | 10.2 | 32.5 |
| o1-mini (5 shot CoT) | 62.3 | 84.5 | 62.1 | 30.6 | 46.5 |
| **Open-source models** | | | | | |
| Llama3-8b (0 shot) | 3.67 | 0.50 | 11.5 | 4.63 | 2.52 |
| Llama3-8b (5 shot CoT) | 20.2 | 17.9 | 17.1 | 12.0 | 28.1 |
| CodeLlama-13b (0 shot) | 1.94 | 0.00 | 8.73 | 1.85 | 0.50 |
| CodeLlama-13b (5 shot CoT) | 9.79 | 8.41 | 13.1 | 9.7 | 9.57 |

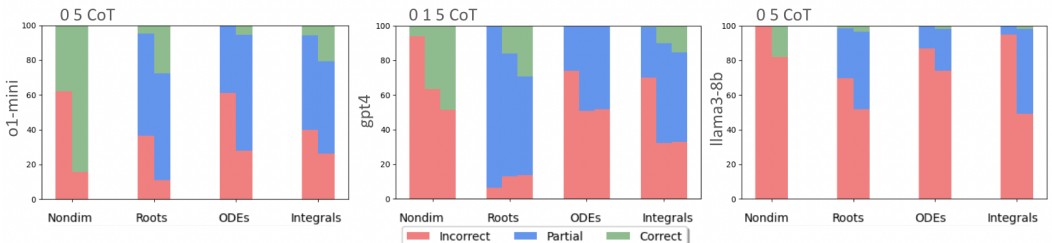

Figure 2: Breakdown of model accuracy percentages for o1-mini, GPT-4 and Llama3 by prompting types and problem types.

Finally, to assess how well LLMs can solve these problems when situated in realistic research contexts, we evaluate GPT-4 (the best performing stable model) on a set of word problems covering all problem types (Appendix A.2). This yields an overall accuracy of 28.1%. Overall, this analysis highlights the value of **HARDMATH** as a challenging benchmark for evaluating mathematical capabilities of LLMs on advanced approximate analytical mathematics.

## 5 Conclusion

We introduce **HARDMATH**, a new dataset covering several problem types from an advanced asymptotics course that can be used to benchmark LLMs' mathematical capabilities and perform model developments. This dataset consists of 1060 examples, and we additionally include 366 verified examples in **HARDMATH-MINI** and 40 verified 'problems in context' that we use to evaluate various leading LLMs. **HARDMATH** is unique as there do not exist large-scale mathematical datasets covering problems of similar difficulty from applied mathematics, and because **HARDMATH**'s problems and solutions are algorithmically generated, meaning that one could produce datasets of arbitrary size using our framework.

Our evaluation highlights that while few-shot CoT prompting significantly improves model performance, especially for models like o1-mini and GPT-4, the overall accuracy on **HARDMATH-MINI** problems remains much lower compared to other existing benchmarks. This suggests that our dataset poses unique and challenging tasks that go beyond the boundaries of current LLM capabilities, particularly in approximation-oriented mathematical reasoning.

Future work will fine-tune LLMs on **HARDMATH** to improve performance. Additionally, while we have evaluated several frontier models, we plan to extend our evaluations to more LLMs as they become available. This expanded evaluation should provide more detailed insights into performance disparities across different models, further advancing our understanding of LLMs' capabilities in handling complex asymptotic reasoning.

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

# A Appendix

## A.1 Implementation and method details for data generation

The following subsections detail the process used to generate the problems and solutions for each problem type.

### A.1.1 Nondimensionalization of polynomials

Nondimensionalization is a technique to simplify equations by reducing the number of parameters [23]. In **HARDMATH**, the first type of polynomial used for nondimensionalization demonstration contains symbolic coefficients and is of the form

$$a_1 x^{n_1} + a_2 x^{n_2} + a_3, \;\; n_1 > n_2 > 0. \tag{1}$$

Nondimensionalization converts this to the form $\epsilon y^{n_1} + y^{n_2} + 1$. The second type contains numerical coefficients and are of the form

$$\pm a_1 x^{n_1} \pm a_2 x^{n_2} \pm a_3, \;\; n_1 > n_2$$

which can be simplified to $\epsilon y^{n_1} \pm y^{n_2} \pm 1$ given a specific *numerical* value of $\epsilon$. Here, integer numerical values for the coefficients $a_1$, $a_2$, $a_3$ are randomly chosen from $[-10, 10]$.

The first nondimensionalization sub-type is generalized by varying the integer values for the degrees $n_1$ and $n_2$ within the range $0 < n_2 < n_1 < 10$, while keeping $a_1, a_2, a_3 > 0$ symbolic. Solutions to these problems express the dimensionless parameter $\epsilon$ in terms of these three coefficients.

---

**Sample Symbolic Nondimensionalization Problem and Full Solution**

**Problem:** Nondimensionalize the polynomial

$$a_1 x^{10} + a_2 x^9 + a_3$$

into one of the form $\epsilon y^{10} + y^9 + 1$. Express $\epsilon$ as a function of $a_1$, $a_2$, and $a_3$.

---

**Solution:** We begin with the substitution

$$x = y \sqrt[9]{\frac{a_3}{a_2}}$$

This gives the expression

$$a_1 y^{10} \left( \frac{a_3}{a_2} \right)^{\frac{10}{9}} + a_3 y^9 + a_3$$

Divide by the coefficient remaining in front of the constant, leaving us with the nondimensionalized polynomial with coefficients in terms of $a_1$, $a_2$, and $a_3$:

$$\boxed{\frac{a_1 y^{10} \left( \frac{a_3}{a_2} \right)^{\frac{10}{9}}}{a_3} + y^9 + 1.}$$

By inspection, we can see that

$$\boxed{\epsilon = \frac{a_1 \left( \frac{a_3}{a_2} \right)^{\frac{10}{9}}}{a_3}.}$$

---

The second subtype implements integer numerical values for the coefficients $a_1$, $a_2$, $a_3$ that are are randomly chosen from $[-10, 10]$.

**Problem:** Nondimensionalize the polynomial

$$P(x) = 2x^7 + 8x^2 + 5$$

into a polynomial of the form $\epsilon y^7 \pm y^2 \pm 1$. Solve for $\epsilon$.

---

**Solution:** For now, we ignore the numeric values of the coefficients and instead call them $a_1, a_2, a_3$. Our polynomial is then:

$$a_1 x^7 + a_2 x^2 + a_3.$$

Use the substitution

$$x = y\sqrt{\frac{a_3}{a_2}},$$

which gives the expression

$$a_1 y^7 \left(\frac{a_3}{a_2}\right)^{\frac{7}{2}} + a_3 y^2 + a_3.$$

Divide all terms by the coefficient remaining in front of the constant term, giving us the nondimensionalized polynomial with coefficients in terms of $a_1, a_2, a_3$:

$$\frac{a_1 y^7 \left(\frac{a_3}{a_2}\right)^{\frac{7}{2}}}{a_3} + y^2 + 1$$

Substituting in the known numeric values for $a_1, a_2, a_3$ (using their absolute values as we have already accounted for sign), we get:

$$\frac{25\sqrt{10} y^7}{1024} + y^2 + 1$$

From inspection of this nondimensionalized equation, we can now identify $\epsilon$:

$$\epsilon = \frac{25\sqrt{10}}{1024} \implies \boxed{\epsilon \approx 0.08.}$$

### A.1.2 Polynomial root-finding

Exact formulas exist for quadratic, cubic, and quartic equations, but deriving them for quintic or higher-order polynomials is not possible [24]. **HARDMATH** includes approximate root-finding examples for higher order polynomials of the form $\epsilon x^{n_1} \pm x^{n_2} \pm 1$. The goal is to solve for roots in terms of $\epsilon$ using the method of dominant balance for small and large positive $\epsilon$ regimes.

As with the nondimensionalization problems, degrees in the polynomial are randomly generated with maximum order ten and $0 < n_2 < n_1$. See a full problem and solution below.

**Problem:** Consider the polynomial

$$P(x) = \epsilon x^6 - x^5 + 1.$$

Find first order approximations for all roots of the polynomials in the limit of small positive $\epsilon$ and large positive $\epsilon$.

---

**Solution:** We begin by equating the polynomial to zero to solve for the roots: $P(x) = 0$. This problem can be rewritten in the form $A + B + C = 0$, where: $A = \epsilon x^6$; $B = -x^5$; $C = 1$.

This problem has no analytical solutions, so we find approximate solutions to the roots by considering the three possible dominant balances. For each dominant balance, we find the

roots of the resulting equation and evaluate whether each balance is self-consistent for small or large positive $\epsilon$.

We start with the balance $A + B = 0$, assuming that $|C|$ is negligible when compared to $|A|$ and $|B|$. Solving this for $x$ in terms of $\epsilon$ then gives us 1 non-zero root:

$$\epsilon x^6 - x^5 = 0$$
$$\implies \boxed{x = \left[\frac{1}{\epsilon}\right].}$$

To verify that these roots are consistent with the assumption that $|A|, |B| \gg |C|$, we substitute these found roots back into the terms $A$, $B$, and $C$ and compare their magnitudes. Using this method, we find that it is true that these roots are valid for small $\epsilon$, while validity for large $\epsilon$ is false.
Therefore, these roots are valid in the limit of small positive $\epsilon$ only.

Next we examine the balance $B + C = 0$, assuming that $|A|$ is negligible when compared to $|B|$ and $|C|$. Solving this for $x$ in terms of $\epsilon$ gives us 5 non-zero roots:

$$1 - x^5 = 0$$
$$\implies \boxed{\begin{aligned} x = &1, \quad -\frac{1}{4} + \frac{\sqrt{5}}{4} - \frac{i\sqrt{2\sqrt{5}+10}}{4}, \quad -\frac{1}{4} + \frac{\sqrt{5}}{4} + \frac{\sqrt{-10-2\sqrt{5}}}{4}, \\ &-\frac{\sqrt{5}}{4} - \frac{1}{4} - \frac{i\sqrt{10-2\sqrt{5}}}{4}, \quad -\frac{\sqrt{5}}{4} - \frac{1}{4} + \frac{i\sqrt{10-2\sqrt{5}}}{4} \end{aligned}}$$

To verify that these roots are consistent with the assumption that $|B|, |C| \gg |A|$, we substitute these found roots back into $A$, $B$, and $C$ and compare their magnitudes. Using this method, we find that it is true that these roots are valid for small $\epsilon$, while validity for large $\epsilon$ is false.
Therefore, these roots are valid in the limit of small positive $\epsilon$ only.

Finally, we examine the balance $A + C = 0$, assuming that $|B|$ is negligible when compared to $|A|$ and $|C|$. Solving this for $x$ in terms of $\epsilon$ gives us 6 non-zero roots:

$$\epsilon x^6 + 1 = 0$$
$$\implies \boxed{\begin{aligned} x = \Bigg[ &-\sqrt[6]{-\frac{1}{\epsilon}}, \quad \sqrt[6]{-\frac{1}{\epsilon}}, \quad \frac{\sqrt[6]{-\frac{1}{\epsilon}}\left(-1-\sqrt{3}i\right)}{2}, \\ &\frac{\sqrt[6]{-\frac{1}{\epsilon}}\left(-1+\sqrt{3}i\right)}{2}, \quad \frac{\sqrt[6]{-\frac{1}{\epsilon}}\left(1-\sqrt{3}i\right)}{2}, \quad \frac{\sqrt[6]{-\frac{1}{\epsilon}}\left(1+\sqrt{3}i\right)}{2} \Bigg]. \end{aligned}}$$

To verify that these roots are consistent with the assumption that $|A|, |C| \gg |B|$, we substitute these found roots back into $A$, $B$, and $C$ and compare their magnitudes. Using this method, we find that it is false that these roots are valid for small $\epsilon$, while validity for large $\epsilon$ is true.
Therefore, these roots are valid in the limit of large positive $\epsilon$ only.

By the Fundamental Theorem of Algebra, a polynomial of degree 6.0 has exactly 6.0 roots. We have found 6.0 roots that are valid in the limit of small positive $\epsilon$ and 6.0 roots valid in the limit of large positive $\epsilon$. Our method therefore provides a complete solution to the problem, finding the correct number of roots in each $\epsilon$ regime.
The roots of $P(x)$ for large positive $\epsilon$ are

$$\boxed{\begin{aligned} &-\sqrt[6]{-\frac{1}{\epsilon}}, \quad \sqrt[6]{-\frac{1}{\epsilon}}, \quad \frac{\sqrt[6]{-\frac{1}{\epsilon}}\left(-1-\sqrt{3}i\right)}{2}, \\ &\frac{\sqrt[6]{-\frac{1}{\epsilon}}\left(-1+\sqrt{3}i\right)}{2}, \quad \frac{\sqrt[6]{-\frac{1}{\epsilon}}\left(1-\sqrt{3}i\right)}{2}, \quad \frac{\sqrt[6]{-\frac{1}{\epsilon}}\left(1+\sqrt{3}i\right)}{2} \end{aligned}}$$

and the roots of $P(x)$ for small positive $\epsilon$ are

$$\frac{1}{\epsilon}, \; 1, \; -\frac{1}{4} + \frac{\sqrt{5}}{4} - \frac{i\sqrt{2\sqrt{5}+10}}{4}, \; -\frac{1}{4} + \frac{\sqrt{5}}{4} + \frac{\sqrt{-10-2\sqrt{5}}}{4},$$
$$-\frac{\sqrt{5}}{4} - \frac{1}{4} - \frac{i\sqrt{10-2\sqrt{5}}}{4}, \; -\frac{\sqrt{5}}{4} - \frac{1}{4} + \frac{i\sqrt{10-2\sqrt{5}}}{4}$$

### A.1.3  Polynomial root correction terms

The use of two-term dominant balances—such as in the previous problem type—neglects terms and introduces an error. We can calculate a correction term $\delta$ to reduce this error via the following: suppose the true roots $x^*$ of a polynomial are given by $x^*(\epsilon) = \overline{x}(\epsilon) + \delta$, where $\overline{x}$ is our approximation to the root (as found in Appendix A.1.2) and $\delta$ is the error term. Plugging the roots $x^*(\epsilon) = \overline{x}(\epsilon) + \delta$ into the polynomial allows one to use a Taylor expansion of $\delta$ around $\overline{x}$ and solve for the correction $\delta$—see the worked solution below.

We also check that resulting solutions have $\delta < \overline{x}$ and exclude solutions that do not meet this criterion.

---

**Sample Numeric Nondimensionalization Problem and Full Solution**

**Problem:** Consider the polynomial

$$P(x) = \epsilon x^3 - x + 1.$$

Find approximate expressions for all roots of the polynomial in the limit of small positive $\epsilon$ and large positive $\epsilon$. Use a series expansion to calculate improved formulae for these roots to order 1 i.e. calculate $\mathcal{O}(1)$ corrections for each root.

---

**Solution:** Note: The root calculation in this problem follow the same method as those demonstrated in the A.3, so they has been omitted here. We include only correction term calculations for the sake of brevity.

We now need to calculate correction terms for these roots to give us better approximations. We consider the ansatz that the root is given by $\overline{x} + \delta$, where the correction term $\delta$ is the sum of higher order terms of $\epsilon$ that we initially neglected in our approximation $\overline{x}$. By definition, $\delta < \overline{x}$. We plug this ansatz into the polynomial and perform a series expansion in $\delta$. We keep terms only up to $\mathcal{O}(1)$ in $\delta$. Then, we set the expression equal to 0 and solve for $\delta$.

Regime 1: valid for small $\epsilon$

Root 1: $-\sqrt{\frac{1}{\epsilon}}$

$$\overline{x} + \delta = -\sqrt{\frac{1}{\epsilon}} + \delta$$

Substitute this into $P(x)$ for $x$ and equate to 0:

$$-\delta + \epsilon\left(\delta - \sqrt{\frac{1}{\epsilon}}\right)^3 + \sqrt{\frac{1}{\epsilon}} + 1 = 0.$$

We then expand this expression to get

$$\delta^3\epsilon - 3\delta^2\epsilon\sqrt{\frac{1}{\epsilon}} + 2\delta - \epsilon\left(\frac{1}{\epsilon}\right)^{\frac{3}{2}} + \sqrt{\frac{1}{\epsilon}} + 1 = 0$$

and represent it as a series of $\mathcal{O}(1)$ in $\delta$, discarding higher order $\delta$ terms

$$2\delta - \epsilon\left(\frac{1}{\epsilon}\right)^{\frac{3}{2}} + \sqrt{\frac{1}{\epsilon}} + 1 \approx 0.$$

We can then solve the expression for the correction $\delta$ to $\mathcal{O}(1)$, and get

$$\delta \approx \frac{\epsilon \left(\frac{1}{\epsilon}\right)^{\frac{3}{2}}}{2} - \frac{\sqrt{\frac{1}{\epsilon}}}{2} - \frac{1}{2}.$$

Root 2: $\sqrt{\frac{1}{\epsilon}}$

$$\overline{x} + \delta = \sqrt{\frac{1}{\epsilon}} + \delta$$

Substitute this into $P(x)$ for $x$ and equate to 0:

$$-\delta + \epsilon \left(\delta + \sqrt{\frac{1}{\epsilon}}\right)^3 - \sqrt{\frac{1}{\epsilon}} + 1 = 0.$$

We then expand this expression to get

$$\delta^3 \epsilon + 3\delta^2 \epsilon \sqrt{\frac{1}{\epsilon}} + 2\delta + \epsilon \left(\frac{1}{\epsilon}\right)^{\frac{3}{2}} - \sqrt{\frac{1}{\epsilon}} + 1 = 0$$

and represent it as a series of $\mathcal{O}(1)$ in $\delta$, discarding higher order $\delta$ terms

$$2\delta + \epsilon \left(\frac{1}{\epsilon}\right)^{\frac{3}{2}} - \sqrt{\frac{1}{\epsilon}} + 1 \approx 0.$$

We can then solve the expression for the correction $\delta$ to $\mathcal{O}(1)$, and get

$$\delta \approx -\frac{\epsilon \left(\frac{1}{\epsilon}\right)^{\frac{3}{2}}}{2} + \frac{\sqrt{\frac{1}{\epsilon}}}{2} - \frac{1}{2}.$$

Regime 2: valid for small $\epsilon$
Root 1: 1

$$\overline{x} + \delta = 1 + \delta$$

Substitute this into $P(x)$ for $x$ and equate to 0:

$$-\delta + \epsilon \left(\delta + 1\right)^3 = 0.$$

We then expand this expression to get

$$\delta^3 \epsilon + 3\delta^2 \epsilon + 3\delta \epsilon - \delta + \epsilon = 0$$

and represent it as a series of $\mathcal{O}(1)$ in $\delta$, discarding higher order $\delta$ terms

$$\delta \left(3\epsilon - 1\right) + \epsilon \approx 0.$$

We can then solve the expression for the correction $\delta$ to $\mathcal{O}(1)$, and get

$$\delta \approx -\frac{\epsilon}{3\epsilon - 1}.$$

Regime 3: valid for large $\epsilon$
Root 1: $\sqrt[3]{-\frac{1}{\epsilon}}$

$$\overline{x} + \delta = \sqrt[3]{-\frac{1}{\epsilon}} + \delta$$

Substitute this into $P(x)$ for $x$ and equate to 0:

$$-\delta + \epsilon \left(\delta + \sqrt[3]{-\frac{1}{\epsilon}}\right)^3 - \sqrt[3]{-\frac{1}{\epsilon}} + 1 = 0.$$

We then expand this expression to get

$$\delta^3\epsilon + 3\delta^2\epsilon\sqrt[3]{-\frac{1}{\epsilon}} + 3\delta\epsilon\left(-\frac{1}{\epsilon}\right)^{\frac{2}{3}} - \delta - \sqrt[3]{-\frac{1}{\epsilon}} = 0$$

and represent it as a series of $\mathcal{O}(1)$ in $\delta$, discarding higher order $\delta$ terms

$$\delta\left(3\epsilon\left(-\frac{1}{\epsilon}\right)^{\frac{2}{3}} - 1\right) - \sqrt[3]{-\frac{1}{\epsilon}} \approx 0.$$

We can then solve the expression for the correction $\delta$ to $\mathcal{O}(1)$, and get

$$\boxed{\delta \approx \frac{\sqrt[3]{-\frac{1}{\epsilon}}}{3\epsilon\left(-\frac{1}{\epsilon}\right)^{\frac{2}{3}} - 1}.}$$

Root 2: $\dfrac{\sqrt[3]{-\frac{1}{\epsilon}}\left(-1-\sqrt{3}i\right)}{2}$

$$\overline{x} + \delta = \frac{\sqrt[3]{-\frac{1}{\epsilon}}\left(-1 - \sqrt{3}i\right)}{2} + \delta$$

Substitute this into $P(x)$ for $x$ and equate to 0:

$$-\delta + \epsilon\left(\delta + \frac{\sqrt[3]{-\frac{1}{\epsilon}}\left(-1 - \sqrt{3}i\right)}{2}\right)^3 - \frac{\sqrt[3]{-\frac{1}{\epsilon}}\left(-1 - \sqrt{3}i\right)}{2} + 1 = 0.$$

We then expand this expression to get

$$\delta^3\epsilon - \frac{3\delta^2\epsilon\sqrt[3]{-\frac{1}{\epsilon}}}{2} - \frac{3\sqrt{3}i\delta^2\epsilon\sqrt[3]{-\frac{1}{\epsilon}}}{2} - \frac{3\delta\epsilon\left(-\frac{1}{\epsilon}\right)^{\frac{2}{3}}}{2}$$
$$+ \frac{3\sqrt{3}i\delta\epsilon\left(-\frac{1}{\epsilon}\right)^{\frac{2}{3}}}{2} - \delta + \frac{\sqrt[3]{-\frac{1}{\epsilon}}}{2} + \frac{\sqrt{3}i\sqrt[3]{-\frac{1}{\epsilon}}}{2} = 0$$

and represent it as a series of $\mathcal{O}(1)$ in $\delta$, discarding higher order $\delta$ terms

$$\delta\left(-\frac{3\epsilon\left(-\frac{1}{\epsilon}\right)^{\frac{2}{3}}}{2} + \frac{3\sqrt{3}i\epsilon\left(-\frac{1}{\epsilon}\right)^{\frac{2}{3}}}{2} - 1\right) + \frac{\sqrt[3]{-\frac{1}{\epsilon}}}{2} + \frac{\sqrt{3}i\sqrt[3]{-\frac{1}{\epsilon}}}{2} \approx 0.$$

We can then solve the expression for the correction $\delta$ to $\mathcal{O}(1)$, and get

$$\boxed{\delta \approx \frac{\sqrt[3]{-\frac{1}{\epsilon}}\left(1 + \sqrt{3}i\right)}{3\epsilon\left(-\frac{1}{\epsilon}\right)^{\frac{2}{3}} - 3\sqrt{3}i\epsilon\left(-\frac{1}{\epsilon}\right)^{\frac{2}{3}} + 2}.}$$

Root 3: $\dfrac{\sqrt[3]{-\frac{1}{\epsilon}}\left(-1+\sqrt{3}i\right)}{2}$

$$\overline{x} + \delta = \frac{\sqrt[3]{-\frac{1}{\epsilon}}\left(-1 + \sqrt{3}i\right)}{2} + \delta$$

Substitute this into $P(x)$ for $x$ and equate to 0:

$$-\delta + \epsilon\left(\delta + \frac{\sqrt[3]{-\frac{1}{\epsilon}}\left(-1 + \sqrt{3}i\right)}{2}\right)^3 - \frac{\sqrt[3]{-\frac{1}{\epsilon}}\left(-1 + \sqrt{3}i\right)}{2} + 1 = 0.$$

We then expand this expression to get

$$\delta^3\epsilon - \frac{3\delta^2\epsilon\sqrt[3]{-\frac{1}{\epsilon}}}{2} + \frac{3\sqrt{3}i\delta^2\epsilon\sqrt[3]{-\frac{1}{\epsilon}}}{2} - \frac{3\delta\epsilon\left(-\frac{1}{\epsilon}\right)^{\frac{2}{3}}}{2}$$

$$- \frac{3\sqrt{3}i\delta\epsilon\left(-\frac{1}{\epsilon}\right)^{\frac{2}{3}}}{2} - \delta + \frac{\sqrt[3]{-\frac{1}{\epsilon}}}{2} - \frac{\sqrt{3}i\sqrt[3]{-\frac{1}{\epsilon}}}{2} = 0$$

and represent it as a series of $\mathcal{O}(1)$ in $\delta$, discarding higher order $\delta$ terms

$$\delta\left(-\frac{3\epsilon\left(-\frac{1}{\epsilon}\right)^{\frac{2}{3}}}{2} - \frac{3\sqrt{3}i\epsilon\left(-\frac{1}{\epsilon}\right)^{\frac{2}{3}}}{2} - 1\right) + \frac{\sqrt[3]{-\frac{1}{\epsilon}}}{2} - \frac{\sqrt{3}i\sqrt[3]{-\frac{1}{\epsilon}}}{2} \approx 0.$$

We can then solve the expression for the correction $\delta$ to $\mathcal{O}(1)$, and get

$$\boxed{\delta \approx \frac{\sqrt[3]{-\frac{1}{\epsilon}}\left(1 - \sqrt{3}i\right)}{3\epsilon\left(-\frac{1}{\epsilon}\right)^{\frac{2}{3}} + 3\sqrt{3}i\epsilon\left(-\frac{1}{\epsilon}\right)^{\frac{2}{3}} + 2}.}$$

### A.1.4 Nonlinear ordinary differential equations

We generate nonlinear third-order ODEs for which there do no exist exact analytical solutions and provide approximate formulae for small and large $x$ regimes, where the small $x$ regime is near $x = 0$ and the large $x$ regime typically involves the solution diverging (example in Appendix A.1.4). The method is robust for higher-order problems, but for simplicity we include only third-order ODEs of the form

$$y''' = f_1(x)(y'')^a + f_2(x)(y')^b + f_3(x)y^c + f_4(x),$$

where $f_1(x), f_2(x), f_3(x), f_4(x)$ are rational functions with integer coefficients. The initial conditions are randomly selected integers from $[0, 3]$. The dataset excludes problems with a function of $x$ as a dominant term because of the difficulty of deriving power law expressions in these cases.

Approximate solutions at small $x$ can be derived using a Taylor series expansion (up to the third order) around $x = 0$. Solving ODEs in the large $x$ regime involves determining the two largest terms, assuming a divergence at some large $x^*$, and solving the dominant balance between these terms to create a power law approximation of the form

$$y(x) = A(x^* - x)^p.$$

---

**ODE Problem and Solution**

Problem: Consider the following third-order ordinary differential equation:

$$y''' = -\frac{y}{24x^4 + 6x^2 + 3} + y'^2 - \frac{y''}{5x^3 - 2x^2 - x + 2} - \frac{1}{12x^2 - \cos(x) + 11}$$

with initial conditions at x = 0:

$$y(0) = 1.00$$
$$y'(0) = 0.00$$
$$y''(0) = 0.00$$

Find analytical expressions that approximate the solution of y(x) at small and large $x$.

---

**Solution:**
The dominant balance in the large x regime is given by

$$\frac{d^3}{dx^3}y = \left(\frac{d}{dx}y\right)^2.$$

We recognize that the solution of this ODE will diverge at finite $x$ and that divergences typically follow a power law of the form

$$y = \alpha(x - x^*)^p,$$

where $x^*$ is the divergence point. The divergence point can be determined by estimated by examining the numerical solution generated by code.

Plugging in the dominant terms we found previously yields the following equation:

$$\alpha p \, (p - 2) \, (p - 1) \, (x - 11.45)^{p-3} = \alpha^2 p^2 \, (x - 11.45)^{2p-2}.$$

After substituting the derivatives, the equation is reorganized to collect terms with respect to $(x - x^*)$. This leads to an equation where the coefficients and powers of $(x - x^*)$ are equated on both sides. Simplifying the equation gives us two separate equations, one for the coefficients and another for the powers of $(x - x^*)$. There is now a system of equations, where the coefficients' equation is

$$\alpha p \, (p - 2) \, (p - 1) = \alpha^2 p^2$$

and the powers' equation is:

$$p - 3 = 2p - 2.$$

Solving this system of equations provides the values of $\alpha$ and $p$. A valid solution is identified if $\alpha$ and $p$ are both nonzero. Here, the solution for $\alpha$ and $p$ is found to be:

$$\alpha = -6, \quad p = -1$$

With these values, the analytical approximation for the solution at large $x$ (near the divergence point) is given by

$$y = -6(x - 11.45)^{-1}.$$

The approximate solution at small $x$ can also be solved used dominant balance, but one can take advantage of the initial conditions and form a Taylor series instead around $x = 0$, which is given by

$$y(x) \approx y(0) + y'(0)x + \frac{y''(0)}{2!}x^2 + \frac{y'''(0)}{3!}x^3.$$

Plugging in the initial conditions, we get the following expression at small $x$:

$$y(x) = 1 - \frac{13}{180}x^3$$

Thus, with rounding for clarity, the solution is given by

$$\boxed{y(x) = 1 - \frac{13}{180}x^3, \ y = -6(x - 11.45)^{-1}.}$$

### A.1.5 Traditional integrals

We consider integrals of the form $I(\epsilon) = \int_0^a \frac{1}{\epsilon + P(x)} \, dx$, where $P(x)$ is an arbitrary polynomial. The dataset provides approximations of each integral in three regimes (small, intermediate, and large $\epsilon$).

The polynomial $P(x)$ is randomly generated to consist of up to ten terms, where each term is a power function of $x$ with an integer power randomly sampled from 1 and 20 and an integer coefficient sampled from 1 to 10. The integration bound $a \in [0, 100]$ is also randomly selected. This form ensures that the integral does not oscillate.

The height is approximated as the maximum value of the integrand, which is $\frac{1}{\epsilon}$, and the width can be estimated as the distance over which the integrand decreases from its maximum value by a factor of 2, which implies that the width $x$ obeys the equation

$$\frac{1}{\epsilon + P(x)} = \frac{1}{2\epsilon} \Rightarrow P(x) = \epsilon.$$

In the regime of small $\epsilon$, the term with the smallest degree and $\epsilon$ are the dominant terms, and in the regime of intermediate $\epsilon$, the term with the largest degree and $\epsilon$ are dominant. There exists one more

solution regime when the width of the integral exceeds the limits of integration, or when $\epsilon$ is "very large." In this case, the integral is approximated by $L/\epsilon$, where $L$ is the integration range.

---

**Sample Integral Problem and Full Solution**

**Problem:**
Consider the integral $I(\epsilon) = \int_0^{56.00} \frac{1}{\epsilon + 2.0x^{6.0} + 2.0x^{9.0} + 5.0x^{11.0} + 5.0x^{13.0}} dx$. Develop analytical formulas that approximate $I(\epsilon)$ for different regimes of $\epsilon$.

---

**Solution:** The integral is of the form $I(\epsilon) = \int_0^{56} \frac{1}{\epsilon + P(x)} dx$ where $P(x)$ is a polynomial. Thus, its value can be estimated as the product between a height and a width.
Since the integrand is maximized at $x = 0$, the height can be set to $\frac{1}{\epsilon}$.
For small $\epsilon$, we define the width as the point where the integrand becomes half of its maximum height. This corresponds to solving for $x$ given $P(x) = \epsilon$. Applying dominant balance, considering the term in $P(x)$ with the smallest degree, the width is approximated as $\left(\frac{1}{2.0*\epsilon}\right)^{1/6.0}$. Therefore, the analytical approximation of the integral for small $\epsilon$ is $I(\epsilon) = \frac{0.8909}{\epsilon^{0.8333}}$.
For an intermediate regime where $\epsilon$ is large, we also define the width based on the term with the largest degree. The width is approximated as $\left(\frac{1}{5.0*\epsilon}\right)^{1/13.0}$. Therefore, the analytical approximation of the integral for large $\epsilon$ is $I(\epsilon) = \frac{0.7647}{\epsilon^{0.8333}}$.
If the width of the integral exceeds the range of integration, we consider one more regime for very large $\epsilon$. The width is then just the range of integration, so in this regime, the integral can be approximated as $\frac{L}{\epsilon}$. Therefore, the analytical approximation of the integral for very large $\epsilon$ is $I(\epsilon) = \frac{56}{\epsilon}$.

Altogether, the solutions at small, large, and very large $\epsilon$ are $\boxed{\frac{0.89}{\epsilon^{0.83}}, \frac{0.76}{\epsilon^{0.83}}, \frac{56}{\epsilon}}$.

---

### A.1.6 Laplace integrals

We consider integrals of the form $I(x) = \int_a^b g(t)e^{\pm x f(t)} dt$, which can be approximated using Laplace's Method when $x$ is very large because the integral's value is dominated by the region around $t_0$ [11].

Laplace integrals of the form $I(x) = \int_a^b g(t)e^{\pm x f(t)} dt$ assume that $f(t) > 0$, is never a constant, and has an absolute minimum at a point $t_0$ either in the interior of or on the bounds of the interval $[a, b]$. Depending on the where the minimum is, the approximate solution is either

$$I(x) \approx g(t_0)e^{\pm x f(t_0)}\sqrt{\frac{2\pi}{x|f''(t_0)|}} \quad \text{or} \quad I(x) \approx \frac{g(t_0)e^{\pm x f(t_0)}}{x|f''(t_0)|}.$$

The set of possible Laplace integrals $I(x)$ in our dataset are parameterized by four parameters: the bounds $[a, b]$, $g(t)$, $f(t)$, and the sign in front of $x$. To generate the dataset, the bounds for each problem were randomly sampled from the $[-1, -0.9, \ldots 0.9, 1]$, and the sign was uniformly sampled from $\{-1, 1\}$. The functions $f(t)$ and $g(t)$ were generated by randomly selecting a linear combination of polynomials up to fifth order and basic trigonometric functions.

Our solution uses SymPy under the hood to find the minima of $f(t)$ (or the dual annealing algorithm if SymPy fails to return the minima).

---

**Laplace Integral Problem and Solution**

Problem: Consider the integral

$$I(x) = \int_{-0.9}^{0.3} (-1.6t^2 - 0.5\sin(t) - 1.9)e^{+x(-2.5t^4 - 0.8t^3 + 1.4t^2)} dt \qquad (2)$$

---

Develop an analytical formula for $I(x)$ that is accurate as $x \to \infty$.

**Solution:**
The integral is of the form

$$I(x) = \int_a^b g(t)e^{+xf(t)}\, dt \tag{3}$$

where $a = -0.9$, $b = 0.3$, $g(t) = -1.6t^2 - 0.5\sin(t) - 1.9$, and $f(t) = -2.5t^4 - 0.8t^3 + 1.4t^2$. This means we can use Laplace's method to develop an analytical approximation in the limit that $x \to \infty$. In this limit, the integral will be dominated by the integrand near the maximum of $f(t)$ within the bounds $[-0.9, 0.3]$. So, to simplify the integral, we will expand the integrand around this maximum. In this case, we can find the maximum of $f(t) = -2.5t^4 - 0.8t^3 + 1.4t^2$ on the interval analytically. We begin by looking for critical point(s) $t_{crit}$ of $f(t)$ by solving $f'(t) = -10.0t^3 - 2.4t^2 + 2.8t = 0$ for $t$. This gives us that $t_{crit} = [-0.66, 0]$. To find the maximum on this interval, we evaluate $f(t)$ at the critical point(s) $t_{crit}$ and the bounds $-0.9$ and $0.3$. We take the $t$ that gives the largest value. Here, this maximum $t_0 = [-0.66]$. Since the integral is dominated by the value of the integrand near -0.66, we Taylor expand the integrand around this point.

$$I(x) = \int_a^b \left( g(-0.66) + (t + 0.66)g'(-0.66) + ... \right)$$
$$* e^{+x(f(-0.66) + (t+0.66)f'(-0.66) + \frac{(t+0.66)^2}{2}f''(-0.66) + ...)}\, dt \tag{4}$$

But $f'(-0.66) = 0$ by definition, so we can remove this term from the exponent. We can then approximate

$$I(x) \approx \int_a^b g(-0.66)e^{+x(f(-0.66) + \frac{(t+0.66)^2}{2}f''(-0.66))}\, dt, \tag{5}$$

which equals

$$g(-0.66)e^{+xf(-0.66)} \int_a^b e^{+x(\frac{(t+0.66)^2}{2}f''(-0.66))}\, dt \tag{6}$$

We perform the change of variables $u = \sqrt{x\frac{|f''(-0.66)|}{2}}(t + 0.66)$, rewriting the integral as

$$g(-0.66)e^{+xf(-0.66)} \int_{\sqrt{x\frac{|f''(-0.66)|}{2}}(a+0.66)}^{\sqrt{x\frac{|f''(-0.66)|}{2}}(b+0.66)} \sqrt{\frac{2}{x|f''(-0.66)|}}e^{-u^2}\, dt \tag{7}$$

Since $x \to \infty$, we approximate this as

$$g(-0.66)e^{+xf(-0.66)} \sqrt{\frac{2}{x|f''(-0.66)|}} \int_{-\infty}^{\infty} e^{-u^2}\, dt \tag{8}$$

Solving the integral and evaluating, we find that

$$\boxed{I(x) \approx -1.21\sqrt{\frac{\pi}{x}}e^{0.37x}} \tag{9}$$

## A.2 Word problems in context

One motivation for creating **HARDMATH** is to help LLMs recognize and solve problems where approximation techniques are needed. To evaluate how LLMs perform on such problems in realistic scenarios, we sample a subset of examples from each problem type and convert these into word problems with a fictional context, creating a dataset of 40 such problems and solutions (see example in the box below). This approach mirrors a format commonly found in textbooks, where a fictional setting provides additional context for the LLM to parse.

Although this dataset is smaller than our hand-verified evaluation set, it is large enough to evaluate the effect of additional context in the problem statement on LLM accuracy.

---

### 2. Sample Word Problem with Context

The density of fish at different points along a certain path in a lake can be modeled as $(\epsilon + x^2 + x^5)^{-1}$, where $x$ represents the distance from the shore in kilometers (ranging from 0 to 100 km), and $\epsilon$ represents environmental factors that affect the fish density. To study the total presence of fish along the path, develop an approximate analytical formula for $I(\epsilon)$ given below:

$$I(\epsilon) = \int_0^{100} \frac{1}{\epsilon + x^2 + x^5} \, dx.$$

## A.3 Evaluation setup

### A.3.1 Prompts for response generation

Table 3: Problem type specific hints by Question and Answer Type

| Question Type | Answer Type | Task instruction |
|---|---|---|
| Nondim-symbolic | SymPy | Please answer the question requiring an answer in a SymPy convertible formula containing variables and math operation expressions and provide the final answer, e.g., $x^3$, $\frac{x}{y}$ inside a Latex boxed format \boxed{}. |
| Nondim-numerical | Float (2) | Please answer the question requiring a floating-point number with two decimal places and provide the final value, e.g., 0.80, 3.12, inside a Latex box \boxed{}. |
| Polynomial Roots | SymPy List | Please answer the question requiring a Python list containing SymPy convertible formulas of variable $\epsilon$ and math operation expressions and provide the final list, e.g., $[\epsilon^3, \frac{1}{\epsilon}]$ inside a Latex boxed format \boxed{}. |
| ODEs | SymPy List | Please answer the question requiring a Python list containing SymPy convertible formula of $y = f(x)$ and provide the final list, e.g., $[y = 1 - x^3, y = -6/(x-5)]$, inside a Latex boxed format \boxed{}. |
| Integrals | SymPy | Please answer the question requiring an answer in a SymPy convertible formula containing formulas of variable $x$ and math operation expressions and provide the final answer, e.g., $x^3$ inside a Latex boxed format \boxed{}. |

### A.3.2 Prompts for grading

Table 4: LLM-based grading prompts by Question and Answer Type

| Question type | Answer type | Task instruction |
|---|---|---|
| Polynomial Roots | SymPy List | Please take this response `{response}` and this answer key `{answer key}` and grade the response based on the following criteria: 1) Check both the small and large $\epsilon$ solutions. 2) For each solution, give full credit if it completely matches the elements in the answer key; give partial credit proportional to the number of matching roots between the response and the answer key; give no credit if it is completely wrong. 3) For both partial and no credit briefly state the error reason. 4) Average the scores for the small and large epsilon solutions to obtain a final score between 0 and 1. 5) Give the final grading as a float in Latex boxed format \boxed{}. |
| ODEs | SymPy List | Please take this response `{response}` and this solution `{answer key}` and grade the response based on the following criteria: 1) Check both the small and large $\epsilon$ solutions. 2) For small regime solution, only give full credit if it matches the formula in the answer key exactly; give no credit if it is doesn't match the form. For large regime solution, give full credit if it matches the formula in the answer key exactly; give partial credit if it doesn't match but the numerical evaluation is not far from solution at this regime; give no credit if neither satisfies 3) Average the scores for the small and large epsilon solutions to obtain a final score between 0 and 1. 4) Give the final grading as a float in Latex boxed format \boxed{}. |
| Integrals (traditional) | SymPy List | Please take this response `{response}` and this solution `{answer key}` and grade the response based on the following criteria: 1) Check both the small and large $\epsilon$ solutions. 2) For each solution, give full credit if it matches the formula in the answer key; give no credit if it is completely wrong and briefly state the reason for the error. 3) Average the scores for the small and large epsilon solutions to obtain a final score between 0 and 1. 4) Give the final grading as a float in Latex boxed format \boxed{}. |
| Integrals (Laplace) | SymPy | Please take this response `{response}` and this solution `{answer key}` and grade the response based on the following criteria: 1) Check the large $x$ final solution. 2) Give full credit if it matches the formula in the answer key; give half credit if the `{response}` get to the checkpoint where it correctly identifies $t_0$ where $f$ attains its maximum and attempt performing Taylor's expansion around it but the final answer is wrong; give no credit if it is completely wrong. 3) For both partial and no credit briefly state the error reason. 4) Give the final grading as a float in Latex boxed format \boxed{}. |

### A.3.3 GPT grading human verification

| Model | Roots | ODEs | Integrals |
|---|---|---|---|
| GPT3.5 (0) | 0 | 0 | 0 |
| GPT3.5 (1) | 0 | -0.09 | -0.02 |
| GPT3.5 (5) | +0.02 | +0.07 | +0.02 |
| GPT4 (0) | 0 | -0.02 | 0 |
| GPT4 (1) | 0 | -0.04 | -0.02 |
| GPT4 (5) | +0.07 | -0.07 | -0.15 |
| o1-mini (0) | +0.04 | +0.05 | 0 |
| o1-mini (5) | +0.05 | +0.05 | 0 |
| Llama3-8b (0) | 0 | 0 | -0.02 |
| Llama3-8b (5) | -0.07 | -0.02 | -0.02 |
| Codellama3-14b (0) | 0 | -0.02 | 0 |
| Codellama3-14b (5) | 0 | -0.02 | 0 |

Table 5: Average adjusted points using human judgment from GPT-based grading. Rows with score adjustments of 0.1 or more are highlighted in pink.

### A.3.4 Model hyper-parameters

Table 6: Generating parameters for various LLMs.

| Model | Generation Setup |
|---|---|
| GPT-3.5 | `model = gpt-3.5-turbo, temperature = 0, max_tokens = 4000` |
| GPT-4 | `model = gpt-4-turbo, temperature = 0, max_tokens = 4000` |
| o1-mini | `model = o1-mini, temperature = 0, max_tokens = 4000` |
| Llama3 | `model = llama3:8b, temperature = 0` |
| CodeLlama | `model = codellama:13b, temperature = 0` |

### A.3.5 Computing resource

Evaluations of open-source models on **HARDMATH** are conducted on a high-performance compute cluster with a single Tesla V100 GPU (16GB vram). Evaluation on one problem type typically takes less than 1 hour. Evaluations of open-source models on **HARDMATH** are conducted on the O2 High Performance Compute Cluster, supported by the Research Computing Group, at Harvard Medical School. See https://it.hms.harvard.edu/our-services/research-computing for more information. Evaluation on one problem type typically takes less than 1 hour with a single Tesla V100 GPU (16GB vram). We would like to thank the HMS Research Computing Consultant Group for their consulting services, which facilitated the computational analyses detailed in this paper.

## A.4 Extended experimental results

### A.4.1 Extended evaluation results

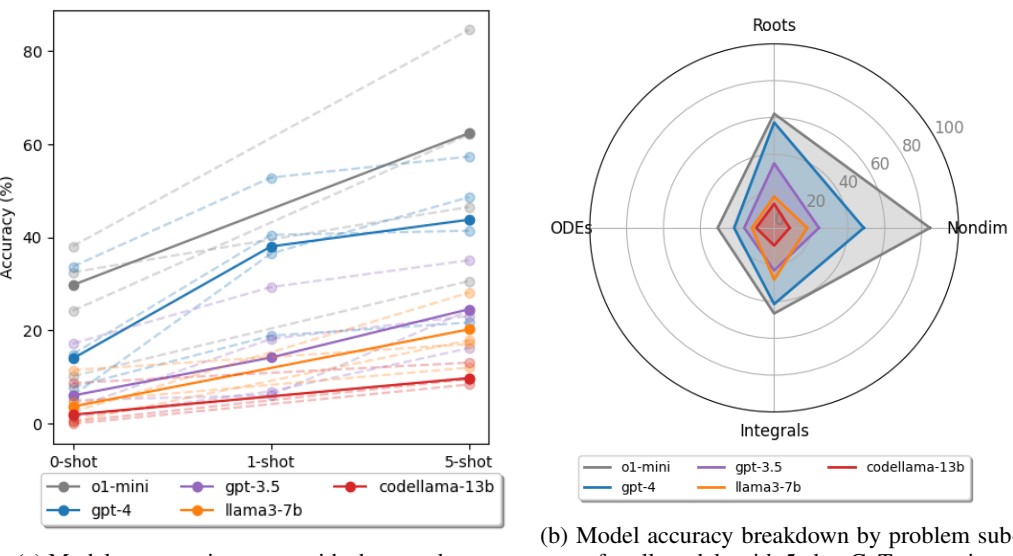

(a) Model accuracy increases with shot numbers.

(b) Model accuracy breakdown by problem sub-types for all models with 5-shot CoT prompting.

Figure 3: Model evaluation accuracy breakdown by shot number and problem sub-types. (a) evaluation accuracy for all models increases with shot numbers for CoT prompting with o1-mini and GPT-4 showing the most obvious improvements; (b) evaluation accuracy breakdown for all models on all problem sub-types under the 5-shot CoT condition.

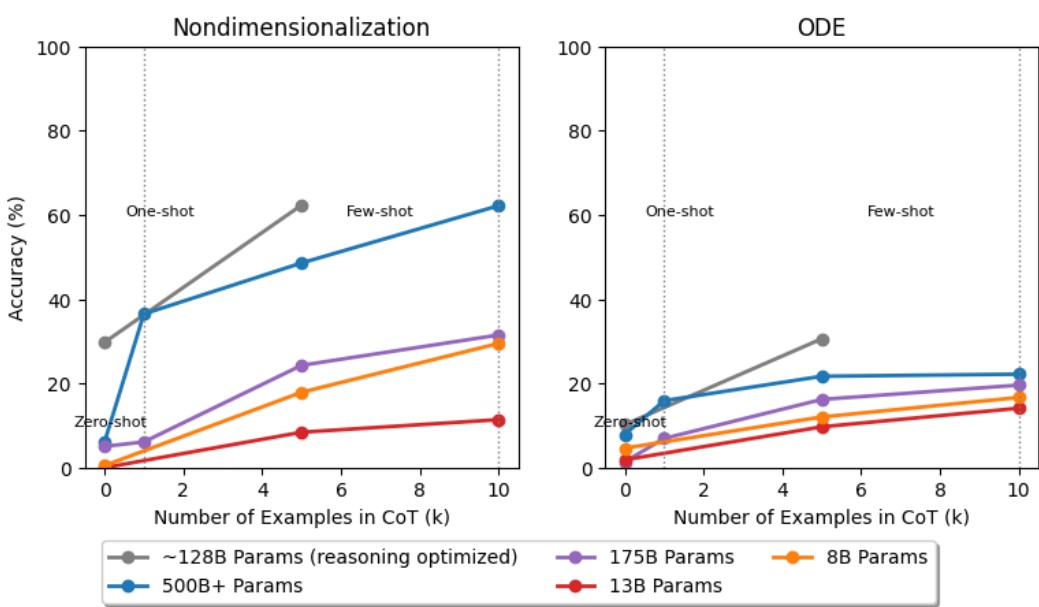

Figure 4: Model performance scaling upon few-shot prompting (0, 1, 5, 10) for problem type *Nondim* and *ODEs*

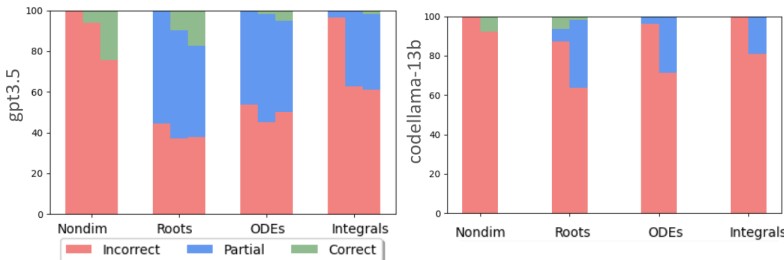

Figure 5: Percentage of correct, partial, and incorrect responses for GPT-3.5 and CodeLlama using CoT.

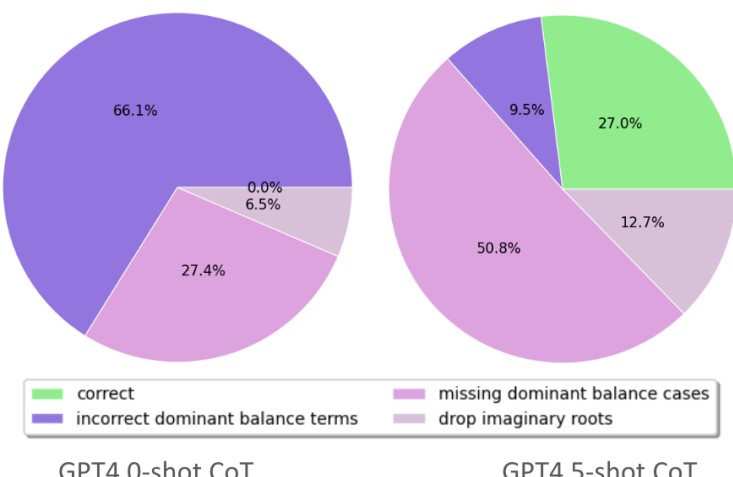

GPT4 0-shot CoT                    GPT4 5-shot CoT

Figure 6: GPT-4 error modes for problem type *Roots* at 0 vs. 5 shot CoT prompting

### A.4.2 Extended error mode analysis

> **Problem 136 Grading Response:** *Roots* (error mode: incorrect dominant balance setup)
>
> **Question:** Consider the polynomial $P(x) = \epsilon x^6 + x^4 - 1$. Find approximate expressions for all roots of the polynomials in the limit of small positive $\epsilon$ and large positive $\epsilon$. Only a single term approximation to the root is required.
> **Grading for Large Positive $\epsilon$:**
> **Model Response:** For large $\epsilon$ When $\epsilon$ is large, the term $\epsilon x^6$ dominates the polynomial for most $x$, except for potentially small $x$ where $x^4$ might still be relevant. However, focusing on the dominant term gives: $P(x) \approx \epsilon x^6$. Setting this equal to zero for an approximation (acknowledging that we're looking for non-trivial roots of the overall polynomial, but focusing on dominance to guide our approximation), suggests that we don't get meaningful roots from this term alone since $x^6 = 0$ doesn't provide roots that satisfy the original polynomial's requirement for non-zero roots.
> **Reference Solution:** Finally, we examine the balance $A + C = 0$, assuming that $|B|$ is negligible when compared to $|A|$ and $|C|$. Solving this for $x$ in terms of $\epsilon$ gives us 6 non-zero roots: $\epsilon x^6 - 1 = 0$ implies $\boxed{x = \left[ \dfrac{(-1 \pm \sqrt{3}i) \sqrt[6]{\frac{1}{\epsilon}}}{2}, \dfrac{(1 \pm \sqrt{3}i) \sqrt[6]{\frac{1}{\epsilon}}}{2}, -\sqrt[6]{\frac{1}{\epsilon}}, \sqrt[6]{\frac{1}{\epsilon}} \right]}$. To verify the roots found for consistency with our initial assumption, we check that $|A|, |C| >> |B|$ holds for this root by substituting our found root expression for $x$ back into $A$, $B$, and $C$ and comparing their magnitudes. Using this method, we find that validity for small $\epsilon$ is False and validity for large $\epsilon$ is True.

---

**Grading**: The response wrongly sets up the dominant balance as $A = 0$ and does not provide a concrete list of roots for large $\epsilon$. Therefore, score for large positive $\epsilon$ is $\boxed{0.0}$

**Problem 133 Grading Response:** *Roots* (error mode: drop imaginary roots)

**Question:** Consider the polynomial $P(x) = \epsilon x^3 + x^2 - 1$. Find approximate expressions for all roots of the polynomials in the limit of small positive $\epsilon$ and large positive $\epsilon$ . Only a single term approximation to the root is required.

**Grading for Large Positive $\epsilon$**:

**Model Response:** To find the approximate expressions for all roots of the polynomial $P(x) = \epsilon x^3 + x^2 - 1$ in the limits of small and large $\epsilon$ , we will consider the two cases separately. For Large $\epsilon$ When $\epsilon$ is large, the term $\epsilon x^3$ dominates the polynomial. Thus, the polynomial can be approximated by ignoring the $x^2$ term, leading to $\epsilon x^3 - 1 \approx 0$ . Solving for $x$ gives us an approximation for the root: $x \approx \left(\frac{1}{\epsilon}\right)^{\frac{1}{3}}$. The other roots in this limit are complex and given the scope, we'll focus on this real root approximation.

**Reference Solution**: Finally, we examine the balance $A + C = 0$ , assuming that $|B|$ is negligible when compared to $|A|$ and $|C|$ . Solving this for $x$ in terms of $\epsilon$ gives us 3 non-zero roots:

$$\epsilon x^3 - 1 = 0 \text{ implies } \boxed{x = \left[ \frac{(-1-\sqrt{3}i)\sqrt[3]{\frac{1}{\epsilon}}}{2}, \frac{(-1+\sqrt{3}i)\sqrt[3]{\frac{1}{\epsilon}}}{2}, \sqrt[3]{\frac{1}{\epsilon}} \right]}. \text{ To verify the}$$

roots found for consistency with our initial assumption, we check that $|A|, |C| >> |B|$ holds for this root by substituting our found root expression for $x$ back into $A$ , $B$ , and $C$ and comparing their magnitudes. Using this method, we find that validity for small $\epsilon$ is False and validity for large $\epsilon$ is True. Therefore, these roots are valid in the limit of large positive $\epsilon$ only.

**Grading**: The response makes additional assumptions to drop the two other complex roots for large $\epsilon$ regime without the question asking it to do so. Therefore, score for large positive $\epsilon$ is $\boxed{0.5}$

### A.4.3   Extended example GPT-4o grading responses

---

**Grading Response:** *ODEs* (incorrect)

To grade the response, we will compare the provided solutions in the response with the ground truth solutions in both the small $x$ and large $x$ regimes.

## Small $x$ Regime

Response: $y(x) = 1 - 5e^{-x}$
Ground Truth: $y(x) = 1 - \frac{5x^3}{6}$
The response and the ground truth solution for small $x$ are significantly different. The ground truth solution is a polynomial in $x$, while the response includes an exponential term. They do not match at all, so the solution for the small $x$ regime is completely wrong.

## Large $x$ Regime

Response: $y(x) =$ "Assumed rapid growth form due to $y'^4$ term."
Ground Truth: $y = \sqrt[3]{30}(x - 2.68)^{\frac{1}{3}} + (1.84)$
The response does not provide a specific analytical expression for the large $x$ regime, while the ground truth does provide a specific form. The response mentions rapid growth due to $y'^4$, but without a clear analytical form, this response doesn't match the ground truth solution. This indicates that the solution for the large $x$ regime is also completely wrong.
Given the evaluation criteria, both solutions do not match or are incomplete compared to the ground truth. Therefore, each solution gets no credit.

The final grading is $\boxed{0.0}$.

Box 2: Example grading response for a *ODEs* problem that gets zero point.

---

### Grading Response: *Integrals* (correct)

The model response approximates the integral for small $\epsilon$ as $\boxed{\dfrac{1}{6^{1/4}}\epsilon^{-3/4}}$ and for large $\epsilon$ as

$\boxed{\dfrac{1}{8^{1/18}}\epsilon^{-17/18}}$.

Now, let's compare these results with the ground truth solutions:

**For small $\epsilon$:** - The ground truth solution is $\boxed{I(\epsilon) = \dfrac{0.64}{\epsilon^{0.75}}}$. - The model's result is

$\boxed{\dfrac{1}{6^{1/4}}\epsilon^{-3/4}}$.

Both results for small $\epsilon$ are essentially the same, with:

$$\frac{1}{6^{1/4}} \approx 0.64$$

**For large $\epsilon$:** - The ground truth solution is $\boxed{I(\epsilon) = \dfrac{0.89}{\epsilon^{0.94}}}$. - The model's result is

$\boxed{\dfrac{1}{8^{1/18}}\epsilon^{-17/18}}$.

Both results for large $\epsilon$ are essentially the same, with:

$$\frac{1}{8^{1/18}} \approx 0.89$$

Since the model's response closely matches the ground truth solutions in both regimes (small and large $\epsilon$) and provides essentially the same approximations, I would give full credit $\boxed{1.0}$

Box 3: Example grading response for a *Integrals* sub-type simple integral problem that gets full points.

---

### Grading Response: *Integrals* (partial)

**Answer Key**
1. Setup and Formulation: it 1) correctly identifies the integral form; 2) uses Laplace's method; 3) identifies critical points and uses Taylor expansion around the point of interest; 4) performs a change of variables to simplify the integral.
2. Key Steps and Approximations: it 1) expands the integrand around the minimum of $f(t)$; 2) approximates the integral by focusing on the dominant exponential term; 3) provides a final analytical approximation for $I(x)$ as $x \to \infty$.
3. Final Result:

$$\boxed{I(x) \approx -\frac{0.004e^{0.89x}}{x}}$$

**Model Answer**
1. Setup and Formulation: it 1) correctly identifies the integral form; 2) recognizes that the exponential term will dominate for large $x$; 3) focuses on the region where the integrand is most significant (near $t = 0.7$).

2. Key Steps and Approximations: it 1) uses the method of steepest descents heuristically without detailed Taylor expansion; 2) approximates by evaluating the polynomial part at $t = 0.7$ and considering the exponential decay; 3) recognizes the dominant behavior of the exponential term but does not perform a detailed variable change or expansion.
3. Final Result:

$$\boxed{I(x) \approx -0.017 \cdot e^{x \cdot 2.6 \cdot 0.7^3}}$$

**Comparison**
1. Similarities: 1) understanding of dominance: both the answer key and the student correctly identify that the behavior of the integral for large $x$ is dominated by the exponential term; 2) identifying significant points: both recognize that $t = 0.7$ is the critical point where the integrand contributes significantly; 3) approximate behavior: Bbth solutions focus on approximating the behavior of the integral for large $x$, emphasizing the exponential term's dominance.
2. Differences: 1) mathematical rigor: the answer key employs a more rigorous mathematical approach, including Taylor expansion and change of variables, providing a more precise and detailed solution; 2) heuristic approach: the student takes a more heuristic approach, focusing on the dominant exponential term without detailed mathematical expansions or variable changes; 3) accuracy of final expression: the answer key provides a more accurate final expression with specific constants derived from the detailed process, while the student's result, though on the right track, has a different coefficient due to the heuristic method. I would give partial credit $\boxed{0.5}$

Box 4: Example grading response for a *Integrals* sub-type Laplace integral problem highlighting the error of failing to develop Taylor's expansion.

