# OpenReview forum: "HARDMATH: A Benchmark Dataset for Challenging Problems in Applied Mathematics"
_NeurIPS.cc/2024/Workshop/MATH-AI — MATH-AI 24_

### Official Review · Reviewer_ypNG · 2024-10-03
**A Review of HARDMATH: A New Benchmark for Evaluating LLMs on Advanced Applied Mathematics**

**Rating:** 8
**Confidence:** 4

**Review:**

**Summary:**

The paper introduces HARDMATH, a novel benchmark dataset designed to evaluate the performance of Large Language Models (LLMs) on advanced applied mathematics problems. These problems are inspired by graduate-level courses and require analytical approximation techniques, which are not well-represented in existing datasets. The dataset aims to fill this gap by providing a challenging testbed for LLMs, focusing on problems that demand a combination of mathematical reasoning, computational tools, and subjective judgment. The authors evaluate several LLMs on a subset of the dataset, demonstrating the limitations of current models in solving these complex problems.

**Quality:**

The paper is well-structured and provides a comprehensive overview of the motivation, design, and evaluation of the HARDMATH dataset. The authors clearly articulate the need for such a dataset and provide a detailed description of the problem types included. The evaluation methodology is robust, employing both automatic and human grading to assess model performance. The results are presented clearly, with appropriate use of figures and tables to illustrate key findings.

**Clarity:**

The paper is generally clear and easy to follow. The introduction effectively sets the stage for the research, and the related work section provides a thorough review of existing datasets and their limitations. The dataset description is detailed, and the evaluation section provides a clear account of the models tested and the results obtained. However, some sections, particularly those involving technical details of the dataset generation and evaluation protocols, could benefit from additional clarification for readers who may not be familiar with the specific mathematical techniques or LLM evaluation methods used.

**Originality:**

The introduction of HARDMATH represents a significant contribution to the field of LLM evaluation. The focus on advanced applied mathematics problems is novel and addresses a clear gap in existing benchmark datasets. The use of a codebase for generating problems and solutions algorithmically is also innovative, allowing for scalability and the creation of datasets of arbitrary size. This approach contrasts with many existing datasets that rely on manual curation and are limited in scope.

**Significance:**

The HARDMATH dataset has the potential to significantly impact the development and evaluation of LLMs in the domain of applied mathematics. By providing a challenging benchmark, it can drive improvements in model capabilities and encourage the development of new techniques for solving complex mathematical problems. The dataset's focus on approximation methods is particularly relevant for scientific and engineering applications, where such techniques are often essential. The paper's findings highlight the current limitations of LLMs in this area, underscoring the importance of continued research and development.

**Pros:**

- **Novelty:** The dataset addresses a previously underrepresented area in LLM evaluation, focusing on advanced applied mathematics problems.
- **Scalability:** The use of a codebase for generating problems and solutions allows for the creation of large-scale datasets.
- **Comprehensive Evaluation:** The paper provides a thorough evaluation of several leading LLMs, highlighting their strengths and weaknesses in solving complex problems.
- **Impact:** The dataset has the potential to drive significant advancements in the mathematical capabilities of LLMs.

**Cons:**

- **Complexity:** Some sections of the paper, particularly those involving technical details, may be challenging for readers without a strong background in applied mathematics or LLM evaluation.
- **Limited Scope of Evaluation:** While the evaluation is comprehensive, it is limited to a subset of the dataset (HARDMath-mini) and a small number of models. Future work could expand this to include more models and the full dataset.
- **Subjectivity in Problem Solving:** The requirement for subjective judgment in solving some problems may introduce variability in evaluation, which could be addressed with more standardized grading rubrics.

---

### Official Review · Reviewer_GAgi · 2024-10-07

**Rating:** 6
**Confidence:** 4

**Review:**

## Paper summary

The paper introduces HARDMATH, a benchmark dataset focused on evaluating LLMs' ability to solve applied mathematics problems that require approximation techniques. Then the authors evaluate several models, including both open-source (e.g., Llama3, CodeLlama) and closed-source models (e.g., GPT-3.5, GPT-4), and identify limitations in current LLM capabilities for advanced applied mathematics.

## Strengths
- It's meaningful to build benchmarks for advanced mathematical reasoning in diverse sub-areas, including applied math.
- The experiments demonstrate limitations in current LLM capabilities.

## Areas to improve
- The name of the dataset is inappropriate. It is too vague and fail to highlight the main feature of the dataset.

- While the authors claim the dataset is designed for applied math, the dataset mainly targets one specific technique in applied math (i.e., the method of dominant balance in asymptotic analysis). Problems that require complex numerical methods (e.g., solving PDEs), stochastic processes, and optimization problems where dominance of terms isn’t the main focus are also important topic in applied math, but the benchmark fails to capture them. I think the authors need to modify the claim in the paper since the dataset does not present a holistic evaluation for applied math capability of LLMs.

- The statistics in Table 1 fails to convince me that the dataset is large-scale. Moreover, the dataset only contains 40 math-word problems, which is very small.

---

### Official Review · Reviewer_E78k · 2024-10-07

**Rating:** 8
**Confidence:** 5

**Review:**

Summary:
The authors introduce a novel dataset for challenging graduate level maths problems and evaluate the performance of SOTA LLMs on their dataset. Authors also provide generation scripts to algorithimically generate problems as well as their corresponding solutions which can be of used for further finetuning.

Pros:
1) Unlike already existing benchmarks JEEBENCH [1] and MATHBENCH [2] which require manual effort for curating problems, the proposed dataset provides generation scripts for algorithmically generating problems
2) Extensive evaluation with multiple open source and closed models has been done showing that there is a significant room for improvement.
3) Extensive error analysis has been done offering insights into complex mathematical reasoning abilities of LLMs showcasing failure modes of LLMs

Minor Points:
1) Apart from standard COT prompting, authors could have experimented with more sophisticated prompting techniques like Show Your Work [3] which would have made the paper more complete.

2) This is not a critcism but just a thought experiment, given the release of o1 by openai which can solve 83% of olympiad level maths questions, It would be interesting to note the performance of o1 on the proposed dataset and look into the corresponding error analysis for the dataset. I hope the authors can include the results with o1 on their dataset when the API becomes publicly available in the future versions of the paper


[1]: Have LLMs Advanced Enough? A Challenging Problem Solving Benchmark For Large Language Models.
[2]: MathBench: Evaluating the Theory and Application Proficiency of LLMs with a Hierarchical Mathematics Benchmark
[3]: Show Your Work: Scratchpads for Intermediate Computation with Language Models

---

### Decision · Program_Chairs · 2024-10-09

Accept